# Evolving Strategies for Extracellular Vesicles as Future Cardiac Therapeutics: From Macro- to Nano-Applications

**DOI:** 10.3390/ijms25116187

**Published:** 2024-06-04

**Authors:** Laura Guerricchio, Lucio Barile, Sveva Bollini

**Affiliations:** 1Biology Unit, Department of Experimental Medicine (DIMES), University of Genova, 16132 Genova, Italy; laura.guerricchio@edu.unige.it; 2Cardiovascular Theranostics, Istituto Cardiocentro Ticino, Laboratories for Translational Research, Ente Ospedaliero Cantonale, CH-6500 Bellinzona, Switzerland; lucio.barile@eoc.ch; 3Euler Institute, Faculty of Biomedical Sciences, Università della Svizzera Italiana, CH-6900 Lugano, Switzerland; 4Cellular Oncology Unit, IRCCS Ospedale Policlinico San Martino, 16132 Genova, Italy

**Keywords:** paracrine effect, extracellular vesicle, heart disease, sustained release, delivery, myocardial targeting

## Abstract

Cardiovascular disease represents the foremost cause of mortality and morbidity worldwide, with a steadily increasing incidence due to the growth of the ageing population. Cardiac dysfunction leading to heart failure may arise from acute myocardial infarction (MI) as well as inflammatory- and cancer-related chronic cardiomyopathy. Despite pharmacological progress, effective cardiac repair represents an unmet clinical need, with heart transplantation being the only option for end-stage heart failure. The functional profiling of the biological activity of extracellular vesicles (EVs) has recently attracted increasing interest in the field of translational research for cardiac regenerative medicine. The cardioprotective and cardioactive potential of human progenitor stem/cell-derived EVs has been reported in several preclinical studies, and EVs have been suggested as promising paracrine therapy candidates for future clinical translation. Nevertheless, some compelling aspects must be properly addressed, including optimizing delivery strategies to meet patient needs and enhancing targeting specificity to the cardiac tissue. Therefore, in this review, we will discuss the most relevant aspects of the therapeutic potential of EVs released by human progenitors for cardiovascular disease, with a specific focus on the strategies that have been recently implemented to improve myocardial targeting and administration routes.

## 1. The Burden of Cardiovascular Disease: From Cell Therapy to Paracrine Strategies

Cardiovascular disease (CVD), including myocardial dysfunction due to cardiomyopathy and ischemic disease up to heart failure (HF), represents a major social and public health burden in Western countries, with high morbidity and mortality rates. Projections estimate that more than 8 million adults will be affected by 2030 [1,2,3]. The adult mammalian heart is known for its very limited reparative potential following severe injury or insult (i.e., acute myocardial infarction—AMI, or cardiotoxicity as a side effect of oncological treatment). Indeed, the myocardial resident functional cell population, namely the contractile cardiomyocytes, shows a very limited turn-over rate during the normal life span [4], with insufficient renewal capacity for therapeutic applications. Following prolonged myocardial ischemia, billions of cardiomyocytes are irreversibly lost [5,6]. The prompt activation of an impaired wound healing response, transitioning from an emergency life-saving mechanism to detrimental remodeling, ventricle dilation, and scarring, may lead to progressive HF [7,8]. HF may also result from cancer-related cardiotoxicity; anthracyclines (including doxorubicin) are a class of oncological drugs that efficiently counteract solid cancers and hematological malignancies. However, they are associated with the early or late onset of off-target chronic cardiotoxicity. Their mechanism of action is still not completely clear, but it is thought to be related to the generation of reactive oxygen species (ROS) and the inhibition of topoisomerase 2β, causing mitochondrial dysfunction and activation of cell death pathways in cardiac cells. Cardiomyocytes are considered the primary cellular targets of anthracycline toxic effects, with progressive development of cardiac dysfunction; yet, cardiac stromal cells, fibroblasts, and endothelial cells may also be affected [9,10,11]. 

Current pharmacological therapies can only delay the onset of myocardial dysfunction and heart failure, with the ultimate solution still being organ transplantation. Against such a scenario, regenerative medicine has suggested alternative strategies based on exploiting the use of progenitor cells in the context of heart repair. Despite initial excitement and high expectations for cell therapy in CVD, a growing body of evidence has shown that progenitor administration results in extremely limited engraftment and meager survival, with a lack of bona fide trans-differentiation [12,13,14]. Nevertheless, several preclinical studies have reported recovery and functional improvement of ischemic or injured tissue, with the consensus pointing to the beneficial effects driven by stem/progenitor cell-secreted paracrine factors [8,15]. This has led to the development of cell-free approaches as proof of principle for future paracrine therapy through the administration of cell secretome formulations, which comprise the entire set of paracrine soluble factors released in the cell-conditioned medium in vitro. For example, patients suffering from ischemic heart disease could benefit from therapeutic interventions enhancing endogenous mechanisms of cardiac repair, including local angiogenesis for collateral vessel formation and muscle preservation and/or regeneration. In this perspective, secretome-based cell-free strategies have been exploited to rescue cardiac function and target the most challenging defects in cardiac disease or following injury, such as limited cardioprotection, exacerbation of inflammation with prolonged activation of fibrosis, and a lack of myocardial renewal. Of note, extracellular vesicles (EVs) released from stem and progenitor cells within their secretomes have gained increasing attention as candidate therapeutic tools, given their role as conveyors of paracrine effects and signal propagators. EVs are heterogenous membrane-surrounded nano-sized particles released by virtually all cells in the extracellular microenvironment as biological conveyors of intercellular communication. The EV content, or cargo, may include various bioactive factors that mirror the state of the parental secreting cell and influence the phenotype and the behavior of the recipient target cell. Notably, this peculiar feature suggests appealing biomarker and therapeutic potential, which has fueled a growing interest in preclinical research over recent years [16]. In this review, we will discuss EV contribution to cardiac repair and regeneration, highlighting their potential as novel therapeutic interventions for cardiovascular diseases. In addition, we will discuss how their ability to target and deliver bioactive molecules directly to damaged or diseased tissues represents an important opportunity for the development of targeted therapies.

## 2. EVs as Biological Conveyors of Cell Communication

EVs are a heterogeneous population of nanosized biological particles characterized by a lipid bilayer [16] and released by different cell types into the extracellular environment. Their content comprises different types of biomolecules, such as proteins, signaling peptides, chemokines, cytokines, lipids, and genetic material including non-coding RNA species (e.g., microRNA or miRNA), collectively referred to as EV cargo. Over the years, EVs have been broadly classified based on their biogenesis into small EVs, also defined as exosomes (30–100 nm in diameter); medium EVs, also indicated as shedding vesicles or microvesicles (50–1000 nm); and large EVs, including migrasomes (500–3000 nm) and apoptotic bodies (50–5000 nm) [17,18]. EV biogenesis involves a series of complex intracellular processes that culminate in their release into the surrounding microenvironment, as summarized in Figure 1. EVs can represent the result of the cellular sorting machinery, including endocytosis, exocytosis, and vesicle-specific intracellular processes such as the endosomal sorting complex required for transport (ESCRT). During endocytosis, cell plasma membranes invaginate to form intracellular vesicles containing material from the extracellular environment. These vesicles can subsequently mature into early and late endosomes, which can be destined for EV formation. Exosomes are formed by the process of membrane invagination in the lumen of the multi-vesicular endosomal compartment (MVE), leading to the formation of small intraluminal vesicles (ILVs) in multi-vesicular bodies (MVBs). These ILVs are released when MVBs fuse with the cell membrane. Exosomes are characterized by the presence of both external and internal proteins, such as tetraspanins (CD63, CD9, CD81), flotillin, annexin, heat shock proteins (HSP70 and HSP90), and biogenesis proteins (tumor sensitive gene and Alix) [19,20]. With respect to exosome biogenesis, the mechanisms of microvesicle formation are still the subject of active research. It is hypothesized that microvesicles are formed mainly through a process of direct protrusion and cleavage from the plasma membrane, mediated by reseptins and other factors that regulate membrane dynamics. However, a complete understanding of these mechanisms requires further investigation [21]. Migrasomes grow at intersections of tubular structures called retraction fibers (RFs) after cell migration, while apoptotic bodies arise from membrane blebbing during cell-programmed death [22]. However, since it is challenging to isolate vesicles of a specific origin through precise separation and concentration techniques, MISEV2023 suggests that the most appropriate classification is based on size, such as small EVs (<200 nm) and large EVs (>200 nm) [16,23]. The rising interest in EVs relies on their crucial role in intercellular communication. Indeed, EVs are conveyors of intercellular communication since they are released into the extracellular space with the advantage of protecting their cargo content from enzymatic degradation [24]. EVs enter body fluids, reach distant tissues, and can (1) link to cell surfaces, initiating intracellular signaling pathways by ligand-receptor interactions; (2) be internalized by recipient cells mainly through clathrin- or caveolin-mediated endocytosis, macro-pinocytosis, and phagocytosis; or (3) enter target cells via direct fusion with the cell membrane, facilitated by proteins such as SNAREs and Rabs [25,26,27,28,29,30,31,32,33]. Nevertheless, the specific mechanism underlying EV uptake from target cells is still a matter of debate [25]. 

The communication mediated by EVs results in the modulation of signaling pathways in the responder cell under physiological conditions as well as during the onset and development of pathological states, depending on their cell source and the molecular mechanisms leading to their biogenesis [31]. EV paracrine effects can range from the activation of responder cells in terms of proliferation, migration, and differentiation, to the modulation of tissue repair mechanisms, and even the alteration of the immune response. By transporting bioactive substances, including proteins, nucleic acids, lipids, and metabolites, EVs mediate the horizontal transfer of molecular instructions and regulatory molecules in the target cells, altering their gene expression and mediating functional effects [32]. Of particular interest, EVs can be sourced in vitro from cells following their release into the cell-conditioned medium (representing the total cell secretome). This has been broadly documented for a wide variety of human cell types, such as mesenchymal stromal cells (MSCs), immune cells, cancer cells, endothelial cells, cardiac cells, and cardiovascular cells. Additionally, EVs can be sourced from biological fluids such as blood plasma, saliva, amniotic fluid, urine, cerebrospinal fluid, and milk [33,34,35,36,37,38,39,40,41,42]. 

## 3. Exploiting EVs in Cardiovascular Disease

EVs may represent interesting tools with different relevant applications in the CVD field. Indeed, depending on the specific cell source, EVs hold great potential for use in the diagnosis and prognosis of cardiac dysfunction, as well as novel therapeutic agents, particularly in the context of acute myocardial infarction and heart failure [43,44]. Their role as diagnostic elements, providing insight into disease onset and the contribution of the different cardiac cells to pathophysiological progression, has been recently demonstrated in different studies [45,46,47,48,49]. Since EVs may play a key role in the myocardial immune response and inflammation, they have been suggested as non-invasive biomarkers for CVD. Circulating EVs, withdrawn from the blood through liquid biopsy, have been profiled based on their microRNA cargo or phenotypic signature as prognostic biomarkers for systemic inflammation with significant clinical impact for SARS-CoV-2 infection [50,51], heart transplant rejection [52], and patient stratification according to the CVD risk profile [53,54] and different stroke types, thus improving diagnosis [55,56,57]. Blood-borne inflammatory EVs have also been demonstrated as representing theranostic targets: they are released in the acute and chronic phases of myocardial ischemia and subsequent inflammatory progression, affecting cardiomyocyte viability and cardiac function [58]. Circulating small EVs in cardiovascular patients may also serve as therapeutic targets for pharmacological modulation. In a recent study, patients with AMI and undergoing coronary angioplasty were infused with cytidine-5’-diphosphocholine at reperfusion and for the following 5 days; plasma-derived EVs showed a different miRNA cardioprotective profile with improved paracrine effects on in vitro cell viability [59].

Progenitor cell-derived EVs have fueled mounting enthusiasm as candidate treatments for the delivery of trophic and stimulatory paracrine effects to diseased myocardium. These EVs can trigger pro-survival, pro-angiogenic, and anti-fibrotic mechanisms, fostering cardiac repair and regeneration. Notably, EVs possess many attractive advantages as innovative cardioactive and cell-mimetic therapeutics, offering a cell-derived, yet cell-free strategy. They exhibit low immunogenicity, biocompatibility, stability, and low cytotoxicity—qualities that are difficult to achieve with artificial nanoparticles [30,60,61,62]. Moreover, EVs may offer unequivocal advantages over canonical cell therapy as a ready-to-use formulation for drug delivery, providing off-the-shelf clinical readiness in combination with low immunogenicity and long-term stability (as extensively reviewed in [63,64]). Several preclinical studies have indicated that human mesenchymal stromal cell (MSC)-derived EVs are an appealing approach to limit cardiac damage and heart disease, overall improving cardiac function [44,65,66,67,68,69], as indicated in the schematic in Figure 2. Differences in the therapeutic use of several types of progenitor-derived EVs may depend on their inherent characteristics, the availability of isolation and characterization methods, and the specific needs of research or therapeutic applications [70].

In this review, we will consider some critical aspects that have yet to be properly addressed for future translation of progenitor cell EVs as candidate therapeutics against CVD. 

## 4. Defining the Optimal Source of EVs for Future Cardiac Paracrine Therapy 

Isolation feasibility, elevated self-renewal, prolonged cryopreservation with a stable karyotype, and a tunable secretory profile are all key aspects of an ideal cell source of cardioactive EVs to be exploited for future paracrine therapy. Different human stem/progenitor cells and MSC populations have been investigated as relevant sources for therapeutic EVs. Indeed, a growing body of evidence suggests that the release of paracrine mediators, including EVs, may represent the bona fide mechanism of action of MSCs in limiting cardiac remodeling, fibrosis, and inflammation while improving cardiomyocyte survival in many preclinical models of CVD. 

*Mesenchymal stromal cells.* Adult MSCs are commonly obtained from bone marrow and adipose tissue samples. Human bone marrow MSC-EVs (hBM-MSC-EVs) have been shown to promote HUVEC proliferation, migration, and tube formation, suggesting a pro-angiogenic effect [81]. This was further confirmed in vivo in a rat model of AMI, where hBM-MSC-EVs injected intramyocardially led to a significant reduction in infarct size while promoting neovascularization and overall enhancing cardiac function. Similar beneficial effects were described by other independent studies, where hBM-MSC-EVs rescued neonatal mouse cardiomyocytes against oxidative damage by reducing cellular damage and apoptosis, as assessed with Annexin V/Propidium Iodide (PI) staining [71]. Human adipose tissue-derived MSC-EVs (hAD-MSC-EVs) have also been tested in vitro on induced pluripotent stem cell-derived cardiomyocytes in an experimental model of cardiac hypertrophy [72]. The administration of hAD-MSC-EVs resulted in reduced protein expression of hypertrophic markers such as the atrial natriuretic factor (ANF), type 1 collagen alpha 1 (COL1A1), and decreased gene expression of the pro-inflammatory interleukin-6 (*IL-6)* [72]. Murine adipose tissue MSC-EVs (AD-MSC-EVs) have been tested on rats with doxorubicin-induced heart failure, leading to improved cardiac function, as shown by increased myocardial ATP content, ejection fraction, and fractional shortening. Serum levels of atrial natriuretic peptide (ANP), which indicate HF progression, were substantially lowered by AD-MSC-EV priming. Additionally, there was a significant decrease in pro-apoptotic markers such as Bax, Caspase-3, and p53 in the cardiac tissue [85]. Likewise, rat AD-MSC-EVs administered to a rat model of AMI counteracted myocardial fibrosis, improved cardiac function, and promoted macrophage skewing towards the pro-resolving M2 phenotype. Serum analysis showed that AD-MSC-EVs reduced AMI-induced levels of IL-6, interleukin-1 beta (IL-1β), interferon gamma (IFN-γ), and tumor necrosis factor-alpha (TNF-α), with a significant contribution from the S1P/SK1/S1PR1 signaling pathway [77]. Rat bone marrow-MSC (BM-MSC-EVs) have also been tested in an in vitro model of cardiac hypertrophy on H9c2 cells, resulting in the downregulation of Bax and Caspase-3 and upregulation of Bcl-2, overall reducing apoptosis. Inflammation was also curbed, as evidenced by decreased levels of brain natriuretic peptide (BNP), IL-1β, IL-4, IL-6, and TNF-α. These cardioprotective effects have been associated with the Hippo-YAP pathway, which regulates cell proliferation and apoptosis in various diseases, including HF [73,74].

Adult human MSCs have shown an excellent safety profile in clinical assessments; however, significant inter-donor variation in terms of yield, limited self-renewal capacity, and phenotypic drift during their in vitro expansion have been also reported. Additionally, the risk of exposure to a lifetime of environmental stimuli (i.e., *inflammaging*) may negatively influence their therapeutic profiles. Therefore, human pre/perinatal MSCs have been recently suggested as an appealing alternative. Extra-embryonic annexes obtained as leftover samples from prenatal diagnosis (such as II trimester amniotic fluid via routine amniocenteses) or clinical waste material at birth (i.e., discarded term placenta membranes and umbilical cord tissue) are enriched in developmentally immature MSCs expressing relevant therapeutic paracrine potential, as indicated by several preclinical models of CVD. Human amniotic fluid-derived stem cells (hAFSCs) have been described as a source of EVs exerting relevant cardioprotective effects in preclinical models of myocardial injury. hAFSC-EVs rescued cardiac function in a rat model of AMI with relevant effects up to 28 days post-injury following a single intramyocardial treatment in the acute setting [82]. Resident cardiomyocytes showed cell-cycle re-entry in the peri-infarcted zone with active incorporation of bromodeoxyuridine, and epicardial progenitor cells reactivated the expression of the key embryonic transcription factor WT1, indicating a reawakening of endogenous regenerative processes. Likewise, hAFSC-EVs delivered intravenously in a rat model of ischemia/reperfusion reduced infarct size [86]. Additionally, anti-fibrotic and pro-angiogenic effects have also been described by hAFSC-EVs as systemically delivered to a rat model of isoproterenol (ISO)-induced cardiac fibrosis. Administration of hAFSC-EVs after 2 weeks from the induction of fibrosis led to lower protein levels of collagen 1 and α-SMA, while promoting angiogenesis and microvascular network [87]. hAFSC-EVs have also been employed to extend the perinatal regenerative window for myocardial renewal in a neonatal mouse model of AMI, both in vitro and in vitro. Mouse neonatal ventricular cardiomyocytes were stimulated with fetal hAFSC-EVs (from amniotic fluid of II trimester gestation from amniocentesis) versus perinatal hAFSC-EVs (as from amniotic fluid obtained at term from scheduled C-section procedures) [83]; only cells treated with the more immature fetal hAFSC-EVs showed a significant increase in the progression from the cell cycle from S- to M-phase up to cytokinesis, with signs of de-differentiation with downregulation of Cofilin-2 (*CFL2),* a miRNA-targeted regulator of cytoskeleton and sarcomere disassembly. Fetal hAFSC-EVs delivered by intraperitoneal injection increased cardiomyocyte cell cycle progression in the 4-day-old neonatal left ventricle myocardium shortly after AMI; yet, this effect was lost at a later stage. Fetal hAFSC-EVs were enriched with an isoform of Agrin, a mediator of neonatal heart regeneration acting through YAP-related signaling. Similarly, human umbilical cord-MSCs-EVs (hUC-MSC-EVs) have been described as delivering cardioprotective effects in limiting cardiac fibrosis and mediating recovery of cardiac function in the long-term when administered intravenously in a rat model of acute AMI [88]. Term placenta membranes represent another appealing perinatal derivative from which mesenchymal stromal cells, and their secreted EVs, can be isolated. Human placental mesenchymal stromal cells-EVs (hPMSCs-EVs) have been shown to decrease plasma and myocardial aspartate transaminase (AST) and brain natriuretic peptide (BNP) and quench local myocardial in inflammation by decreasing pro-inflammatory interleukins IL-1β, IL-6 and tumor necrosis factor-α (TNF-α), overall limiting tissue injury in a mouse model of AMI when administered intravenously [78].

*Cardiac progenitor stromal cells.* Remarkable cardioprotective effects have also been observed from EVs released from endogenous cardiac cell populations within the heart. Cardiac progenitor/stromal cells (CPCs) represent a rare population with relevant potential to contribute to cardiogenesis during embryonic development, which becomes almost completely quiescent in adulthood. While the role of CPCs in actively contributing to cardiovascular lineages postnatally and following injury is currently a matter of debate [89,90,91], several lines of investigation have highlighted their relevant trophic paracrine effects [92,93]. Epicardium-derived progenitor cells (EPDCs) can release EVs enhancing the proliferation of neonatal murine cardiomyocytes in vitro. When injected into the injured area of infarcted neonatal hearts, EPDCs promoted cell cycle re-entry via the activation of Akt, Hippo, and ERK pathways [84]. Human EPDC-EVs were also tested in vitro on an engineered human myocardium (EHM) cryoinjury model, which mimics cardiomyocyte loss and decreased force generation. At day 7 post-injury and treatment with EPDC-EVs, the contractile function was enhanced. Both murine and human EPDC-EVs were found enriched with cardioprotective miRNAs, including *miR-99a-5p, miR-30e-3p, miR-30a-5p, miR-21-5p, miR-23b-3p, miR-181a-5p, miR-27a-3p, miR-100-5p*, and *miR-146a-5p*. Human CPCs (hCPCs), obtained from atrial appendage explants from patients who underwent heart valve surgery, have been shown to improve cardiac function after injury. The EVs released from such cardiac stromal populations have been recently broadly investigated in several preclinical models of myocardial injury, including AMI and pharmacologically-induced cardiotoxicity. EVs were shown to be the cardioprotective component of the paracrine secretion of hCPC, both in vitro and in vivo. hCPC-EVs inhibited HL-1 cardiomyocyte cell death, likely due to their *miR-210* content targeting pro-apoptotic ephrin A3 and PTP1b, while supporting tube formation in HUVECs via *miR-132* delivery. When injected into infarcted rat hearts, they counteracted resident cardiomyocyte apoptosis, enhanced cardiac function, and supported local angiogenesis [75]. Similar results were validated in a large animal model, wherein intracoronary delivery of hCPC-EV reduced the infarct size in porcine acute myocardial infarction [94,95]. hCPC-EVs were also tested in preclinical murine models of doxorubicin/trastuzumab-induced cardiotoxicity. Proteomic profiling of hCPC-EVs has indicated an enrichment of proteins involved in redox processes. When systemically injected in a rat model of oncological therapy-derived cardiotoxicity, they significantly contributed to reducing ROS levels in the injured heart, while counteracting fibrosis, decreasing the levels of interstitial collagen 1 deposition and inflammation, and lowering the CD68+ macrophage infiltrate [76]. hCPC-EVs have also been described as highly enriched in *miR-146a,* which was shown to functionally concur with the mechanism of cardioprotective actions exerted on target cells [76,96]. Furthermore, miR-181b encapsulated EVs derived from human cardiosphere-derived cells (CDCs), which are obtained by culturing self-assembling spherical aggregates of hCPC, reduce PKCδ expression in activated monocytes and enhance the macrophage-mediated cardioprotective effects [79].

*Induced pluripotent stem cells*. Another appealing cell source of EVs is represented by induced pluripotent stem cells (iPSCs). Indeed, human iPSCs (hiPSCs) have been used to obtain cardiac and cardiovascular progenitors for cell therapy against heart injury, with encouraging results in terms of cardiac function improvement. These improvements are most likely attributed to the soluble factors secreted by the cells [80,97,98]. Therefore, EVs from iPSC and their derivatives have lately attracted increasing interest in cardiac paracrine therapy. Murine iPSC injected intramyocardially at 48 h after reperfused myocardial infarction in mice showed improved cardiac function. Additionally, murine iPSC-EVs exhibited even greater cardiac repair potential and proved to be safe, whereas iPSC delivery resulted in teratoma development in vivo [99]. EVs obtained from the secretome of human iPSC-derived cardiovascular progenitors (hiPSC-Pgs) have been shown to impart cardioprotective effects on cardiac cells in a mouse model of chronic heart failure. This was achieved by preserving left ventricular function following transcutaneous echo-guided injection in the pre-infarcted zone 3 weeks after AMI [100]. Similarly, hiPSC-Pg-EVs were also tested in a murine AMI model showing a reduction in inflammation. Specifically, there was a decrease in pro-inflammatory M1 macrophages along with an increase in pro-resolving M2 macrophages within the treated myocardium [80]. Furthermore, hiPSC-Pg-EVs reduced levels of pro-inflammatory cytokines, such as IL-1α, IL-2, and IL-6, and simultaneously increased levels of anti-inflammatory cytokine IL-10 [96]. iPSC-cardiac progenitor cell EVs have also been applied to rodent models of chemotherapy-induced cardiomyopathy with intraperitoneal delivery. These EVs contributed to preventing maladaptive remodeling, inhibiting the onset of fibrosis, and decreasing the expression of heart failure molecular indicators (such as myosin heavy chain isoforms *Myh6/Myh7* ratio) [97].

Overall, progenitor cell-derived EVs have demonstrated relevant cardiac repair and regenerative potential by exerting remarkable paracrine modulation, acting on multiple levels on cardiac and cardiovascular cells. These include anti-oxidant, anti-inflammatory, anti-fibrotic, pro-survival, pro-angiogenic, and proliferative effects (as illustrated in Figure 2 and summarized in Table 1). These actions collectively antagonize maladaptive ventricular remodeling and myocardial dysfunction, ultimately leading to cardiomyopathy and heart failure. 

Different human cell sources may offer distinctive advantages, according to the ease of isolation, self-renewal capacity, stability, and the EV yield from their secretome. The ideal source choice should be pondered according to the EV-derived cardioactive or cardioprotective effect for the specific cardiovascular disease (i.e., acute AMI, chronic cardiomyopathy, cardiotoxicity).

## 5. Optimization of Cardiac Delivery: From Macro- to Nano-Applications 

EVs have been shown to deliver pleiotropic beneficial effects on the cardiac tissue by acting on several key mechanisms that characterize the damaged heart, ranging from chronic inflammation and fibrotic remodeling to lack of myocardial regeneration and local angiogenesis, all of which affect cardiac function. Most studies are based on systemic delivery of EVs as it is a less invasive and more clinically relevant administration route. Yet, this strategy may be significantly affected by unspecific uptake by off-target cells of the administered EVs before reaching the myocardial tissue. On the other hand, local intramyocardial injection during cardiac surgery or trans-endocardial delivery using an advanced multicomponent catheter combined with imaging mapping systems may not be a feasible option for all CVD patients, especially when multiple treatment administrations are required. Moreover, once EVs are in the myocardial tissue, they can interact with different resident cardiac and cardiovascular cells, including endothelial, smooth muscle, and epicardial cells, in addition to the typical targets such as cardiomyocytes and fibroblasts. Thus, in order to define a feasible approach for future clinical translation, it is necessary to optimize cardiac delivery strategies to ensure the following: (i) controlled and sustained administration of the EV therapeutic dosage and (ii) cardiac-specific targeting of the EV formulation. The implementation of heart-targeted therapies to boost EV uptake by target cells will dramatically improve paracrine treatment efficiency and effectiveness, while drastically limiting any side effects on other organs, as illustrated in Figure 3. A relevant technical and clinical challenge is also represented by the ability to efficiently deliver ready-to-use EV therapeutics to the myocardium at the appropriate timing and according to the specific CVD situation. Indeed, to prevent detrimental pathological remodeling after MI, paracrine medicinal products should be provided immediately at reperfusion as a single administration during percutaneous coronary intervention. To inhibit long-term and off-target cardiotoxicity of drug treatments (i.e., oncological chemotherapy), paracrine therapeutics should be used in advance or during the concomitant pharmacological therapy through multiple administrations to shield the heart from detrimental side effects. Both scenarios compel the administration route to be optimized to be as patient-compliant as possible. As a matter of fact, the functional validation of EVs requires the design of targeted delivery systems to ensure prolonged paracrine activity and refine the administration regime. 

*Biomaterial-embedding EVs.* The recent evolution of smart biomaterial engineering has led to improved formulations of biocompatible scaffolds that can be combined with EVs to enhance their release into the heart. These formulations offer options for acute single local administration for AMI or controlled release to provide repeated treatments for chronic CVD. Injectable biodegradable hydrogels have demonstrated mechanical protective properties that are beneficial for the ischemic myocardium [101]. Several natural hydrogels have been suggested for cardiac tissue engineering approaches, including fibrin and alginate, collagen, hyaluronic acid (HA), Matrigel, or chitosan-based formulations [102]. Hydrogels with features mimicking the cardiac microenvironment have been designed as EV delivery systems to enhance local intramyocardial retention, maximize paracrine action, and provide medium- to long-term sustained release for AMI patients [103]. Rodent BM MSC-EVs incorporated in alginate hydrogel and injected intramyocardially in the rat infarcted area have been shown to decrease cardiac cell apoptosis, promote macrophage polarization, and improve long-term cardiac function. Hydrogel-embedded EVs were retained in situ, thus ensuring controlled release and preventing their fast diffusion out of the heart. The hydrogel acted as a temporary local reservoir, enabling slow administration and effectively mediating cardiac repair. As a result, the EVs were highly sustained in the heart and scarcely present in the liver, lungs, and spleen, in contrast to the delivery of EVs in free form [104]. More recently, human umbilical cord-MSC-EVs loaded on a clinical-grade HA biomaterial were tested in a clinically relevant model of rodent chronic heart failure. The model mirrors the situation of patients who, despite early reperfusion after myocardial ischemia, subsequently develop detrimental left ventricular remodeling and might require long-term treatment. HA-carried EVs preserved cardiac function, improved angiogenesis, and decreased both apoptosis and fibrosis when compared to intramyocardial administration of EVs in their free form [105]. In situ cardiac patch formation following intrapericardial injection of MSC-EV-loaded biocompatible methacrylic anhydride–HA hydrogels has been recently described. Such a strategy resulted in the hydrogel providing a repository structure in the pericardial cavity, while modulating immune response and increasing the cardiac retention of the therapeutics acting on epicardial progenitors underneath, thus supporting endogenous cardiac repair mechanisms [106]. Similar results have also been obtained in larger preclinical animal models. EVs from porcine cardiac adipose tissue-derived MSCs (cATMSCs) embedded into a pericardial peptide hydrogel scaffold have been delivered to the ischemic myocardium in a pig AMI model. This delivery strategy resulted in the confined administration of the nanoparticles, guaranteeing the local release of the EV dosage with the generation of a vascularized niche for endogenous cell recruitment and modulation of short-term post-ischemic inflammation [107,108]. An injection-free approach to deliver EVs onto the cardiac surface to treat a preclinical mouse model of AMI has also been proposed. This method involves applying a spray mixture of EVs, gelatin methacryloyl (GelMA) precursors, and photo-initiators with visible light irradiation for 30s. The EVs were trapped in the hydrogel network formed in situ and then gradually released with diffusion and enzymatic degradation of the polymer, concurring to improve cardiac function [109]. While this strategy demonstrated that needle-based intramyocardial injection of the EV-loaded smart biomaterial is not necessary, it still requires direct access to the cardiac surface via an invasive approach. 

*Engineering EVs.* These studies support the hypothesis that the progressive release of EVs sustains therapeutic effects while limiting undesirable biodistribution in the body. Nevertheless, not all CVD patients (i.e., patients suffering from cancer-related cardiotoxicity and cardiomyopathy) may be eligible for EV paracrine therapy involving in situ delivery procedures, as required by hydrogel-based systems. Simultaneously, EVs delivered intravenously are rapidly cleared, primarily accumulating in the liver, thus failing to reach the heart and increasing the likelihood of off-target effects. Thus, alternative strategies have been developed to improve the on-target binding of EVs at both cell-type-specific (i.e., cardiomyocytes versus fibroblasts as target cells) and tissue-specific levels through a less invasive systemic injection route. Such strategies focus on nano-functionalization or bioengineering of the EV surface, achievable by modifying the donor secreting cell (MSC, CPC, iPSC, etc.) or directly functionalizing EVs post-purification [110]. EVs can be engineered to express peptides on their surface to increase targeting to the heart tissue and their retention by cardiac cells. Several studies have identified cardiac-specific peptides with different motifs, such as CSTSMLKAC (cardiac homing peptide, CHP), CKPGTSSYC, and CPDRSVNNC [111], exhibiting preferential binding to rat ischemic heart tissue. APWHLSSQYSRT (termed Cardiac Targeting Peptide, CTP) has been shown to be promptly internalized by H9c2 cells in vitro [112], while WLSEAGPVVTVRALRGTGSW (CardioMyocyte specific Peptide, CMP) was found to address a region in an extracellular matrix protein, tenascin-X, preserved across species and specifically for primary cardiomyocytes [113]. To induce the expression of these guiding peptides on the EV surface, the secreting parental cells can be genetically modified in vitro with a lentivirus construct carrying the Lamp2b protein fused to a myocardium-targeting peptide [114]. Cardiosphere-derived cell-derived EVs (CDC-EVs) have been engineered to insert the CMP peptide on their surface using a lentiviral vector encoding for the CMP peptide fused to the Lamp-2b protein [115]. Functionalized CDC-EVs exhibited increased uptake by primary neonatal mouse cardiomyocytes compared to non-engineered CDC-EVs in the short-term post-administration. Ex-vivo imaging analysis further confirmed that CMP-targeted CDC-EVs presented improved retention within the heart, compared to naïve CDC-EVs. The same strategy was used to expose CTP on human HEK293 cell-derived EVs [116], with higher efficiency of uptake from the cardiomyocyte-like H9c2 cell line compared to control EVs only expressing Lamp-2b. In vivo, following injection in the tail vein, CTP-modified EVs were internalized by the cardiac tissue with a 15% increase. Considering that the injured infarcted area may be enriched with cardiomyocyte-released cardiac troponin I (cTnI), the Lamp-2b technology has also been successfully used to express cTnI-targeting-peptide on EVs [117]. A similar approach was performed using the C1C2 domain of the human lactadherin protein instead of Lamp-2b; HEK cells were transfected with a lentiviral plasmid with 3 CHP coding sequences fused to the N-terminus of the C1C2 domain, and intravenous administration of CHP-exposing EVs led to enhanced localization to the rat ischemic myocardium [118].

Modulation of EV biodistribution and targeting can also be achieved through chemical functionalization of the vesicle surface using either copper-free click chemistry, to insert linkers to functional carboxyl or amine groups binding specific peptides on the EV, or by means of physical incorporation/absorption of lipoproteins (as reviewed in [110]). In click chemistry, two molecular structures are joined together via the cycloaddition of an azide to an alkyne molecule to form a triazole with high yield and selectivity [119,120]. MSC-EVs have been functionalized with an alkyne (dibenzylcyclooctyne-sulfo-N-hydroxysuccinimidyl ester, DBCO-sulfo-NHS) that reacts with an azide linked to the CHP peptide [121]. In a mouse preclinical model of AMI, intravenous administration of engineered EVs resulted in a higher and more specific uptake of CHP-MSC-EVs by the ischemic myocardium compared to scramble (Scr)-MSC-EVs, which were mainly retained by the liver, lungs, and kidneys. Additionally, the CTP peptide has been bound to the surface of human peripheral blood-derived EVs (hPB-EVs) by bio-orthogonal copper-free click chemistry to create CTP-hPB-EVs. This modification enhanced the internalization of CTP-hPB-EVs by hiPSC-cardiomyocytes and cardiac AC16 and H9c2 cell lines, compared to EVs modified with scramble peptide (Scr-hPB-EVs). Furthermore, CTP-hPB-EVs demonstrated a high tropism for cardiomyocytes compared to fibroblast and endothelial cells [122]. Functionalization of CPC-EVs with CHP has also been recently reported using a molecular linker without click chemistry. EVs were conjugated with CHP through a dioleoylphosphatidylethanolamine N-hydroxysuccinimide (DOPE-NHS) linker, resulting in increased retention within the myocardial tissue in a mouse model of ischemia/reperfusion injury and significant improvement in cardiac repair outcomes. Enhanced targeting of EVs to injured tissue has further been reported by optimizing an EV membrane anchoring platform known as "cloaking", which involves directly embedding tissue-specific antibodies or homing peptides on the vesicle membrane for enhanced uptake by the target cells. The cloaking system consists of a 1,2-bis(dimethylphosphino)ethane (DMPE) phospholipid membrane anchor, polyethylene glycol (PEG) spacer, and a conjugated streptavidin platform molecule, allowing for the binding of any biotinylated molecule to obtain EV nano-functionalization. This approach results in increased uptake of EVs in cardiac tissue in a rat model of ischemia/reperfusion [123]. 

*Microenvironment mimetics.* Hybrid EVs as enveloped with cell membranes from monocytes and platelets have also been tested in order to take advantage of mononuclear phagocyte system (MPS) evasion, thus lowering EV clearance by macrophages, while exploiting the recruitment feature of monocytes and platelets to injured myocardium. The chemokine ligand CCL2 targets cardiomyocytes, fibroblasts, and endothelial cells and interacts with the corresponding C-C chemokine receptor 2 (CCR2) on the surface of infiltrating macrophages. Recently, cardiac-resident macrophage-EVs (mEVs) have been modified with monocyte membranes, creating CCR2 positive-MmEVs. Enveloping mEVs with monocyte membranes (Ms) resulted in increased EV uptake by cardiac cells and better targeting of the damaged myocardium. This was achieved with MPS evasion and the binding of CCR2-positive MmEVs to CCL2 expressed on cardiac cells in the infarct area. MPS evasion was possible due to the MmEV surface expression of the differentiation cluster 47 (CD47-“*do not eat me*” signal), which interacts with the receptor signal-regulated protein α (SIRP α) on innate immune cells, hence avoiding immune clearance by the MPS and allowing more time for tissue targeting. Both immunofluorescence and flow cytometry analyses revealed that MmEVs were significantly taken up by cardiomyocytes, with a 2-fold increase compared to naïve mEVs. When administered intravenously via the rodent tail vein, MmEVs showed higher retention within the heart with lower accumulation in the liver [124]. A monocyte-mimicking method has also been reported based on the hypothesis that modification of EVs with monocyte membranes could enhance their cardiac retention through interaction with ischemia-injured endothelial cells and cardiomyocytes. Monocyte membranes were prepared from RAW264.7 cells and rat BM-MSC-EVs were enveloped within them (Mon-EVs) using an extrusion method. When injected intravenously in a mouse model of AMI, Mon-EVs showed increased retention in the heart compared to naive EVs, based on the interaction of adhesive molecules, such as the macrophage receptor 1 (Mac1) on the monocyte surface with ICAM-1, which is overexpressed in the infarcted myocardium [125]. 

Another explored biomimetic approach takes advantage of the natural infarct-homing ability of the platelet membrane [126]. This strategy involves creating membrane-camouflaged EVs endowed with a “*do not eat me*” signal via CD47 surface expression to avoid uptake by macrophages. MSC-EVs covered by platelet membrane (P-MSC-EVs) have been shown to significantly accumulate in the injured heart compared to unmodified MSC-EVs, with lower distribution to the liver and kidneys [127]. Another study illustrated that platelet-covered EVs can adhere to the injured vascular wall via the platelet surface glycoprotein (GPIbα) and von Willebrand Factor (vWF) secreted from an activated endothelium, suggesting that targeting may be enhanced by interaction with an injured endothelium [128]. Finally, through the overexpression of the CXC motif chemokine receptor 4 (CXCR4), researchers enhanced the effectiveness of systemic injection of cardioprotective hCPC-EVs by increasing their bioavailability to ischemic hearts, where the release of the natural ligand SDF-1 is enhanced. Intravenous administration of CXCR4-overexpressing EVs notably reduced infarct size and improved left ventricle ejection fraction after 4 weeks compared to naive EVs [129].

## 6. Translational Challenges

To define EVs as innovative investigational drug formulations and advance their potential for future clinical translation, several challenges (including methodological ones) have yet to be properly addressed. Hurdles exist at both basic biology and clinical application levels, particularly in terms of EV separation from non-vesicular extracellular particles within the cell secretome, their characterization, and potency assay studies. Regular updates on available approaches and optimized techniques are provided by the International Society for Extracellular Vesicle (ISEV) position papers, such as the latest version of the “Minimal Information for Studies of Extracellular Vesicles” [16]. 

There is currently no consensus on the ideal protocol for EV purification to ensure yield maximization from stem/progenitor cell secretomes. Indeed, each protocol may present some advantages and critical drawbacks. Most EV separation and concentration methods are based on their physical and chemical features such as size, density, or surface markers. Differential ultracentrifugation (dUC) is one of the most commonly used techniques in the literature. It separates EVs within a solution according to their size and density by applying different acceleration rates (in g or rpm) but may result in EV aggregation with significant protein co-isolation [130,131]. Density gradient ultracentrifugation is based on generating layers with increasing density from top to bottom or in reverse (using sucrose, iodixanol or iodhexol, and aqueous buffers), and the material containing the EVs can be loaded above or below such a gradient. The EVs are collected in a layer with comparable density. This method can be used to separate EVs from proteins in the same preparation, but it may lead to low recovery of high-purity EVs [132,133]. Size exclusion chromatography (SEC) is an easy-to-use technique that separates EVs from contaminating particles, such as lipoproteins, based on their size. This is achieved through a column with a pore-containing matrix, which facilitates the elution of bigger particles first (that do not enter the pores), followed by the elution of the smaller ones [134]. Yet, additional steps to concentrate the starting material before and after SEC may be necessary and time-consuming on the volume to load on the column and the final eluted volume [135,136]. Ultrafiltration is another size-based technique that involves forcing the EV-containing sample through membranes with a specific molecular weight cut-off (mostly from 10 to 1000 kDa) inserted into a column. This strategy includes both dead-end filtration (DF) and tangential flow filtration (TFF), both allowing the separation of EVs (in the retentate) from contaminants (in the permeate) [137]. TFF allows high-volume processing but requires specialized and costly equipment. Moreover, material adhering to the membrane can cause clogging, thus reducing the efficiency and effectiveness of EV separation in both techniques [138,139,140,141,142]. Fluid-flow-based separation (FFS) is a group of methods for the liquid-phase separation of macromolecules or particles in the 1–100-micron range, which is also applied to EVs. In field-flow fractionation (FFF), the sample is separated inside a flow channel, which is a column devoid of a stationary or solid phase, allowing high particle recovery. This technique allows a gentle flow for EV separation and size-based separation with high resolution but is associated with cross-contamination between different fractions and limited scalability due to the small amount of sample that can be loaded [143,144]. Standardization of EV separation and concentration also requires optimization of in vitro culture and expansion of the secreting progenitor cell source. Indeed, large-scale production of Good Manufacturing Practice (GMP)-grade EVs should minimize the risk of batch-to-batch variation. In such a perspective, an in vitro 3D culture of MSCs and their immortalization have been proposed as relevant approaches to boost and standardize EV yield while also improving their therapeutic effects [145,146].

EV heterogeneity represents another important aspect. Cell-derived EVs contain a variety of bioactive paracrine molecules, including proteins, nucleic acids, and lipids, mirroring the parental cell condition, which can eventually be influenced by the separation and concentration method. Since EV heterogeneity may influence their paracrine potential, it represents a critical aspect for the standardization of future therapies. The challenging task ahead involves precisely defining the therapeutic cargo and dosage. Thus, further investigations on reliable potency assays on target cells are necessary to pinpoint the dosage and the kinetics of the cardioactive effects exerted by EVs. Indeed, a better understanding of the impact of EV heterogeneity in clinical application is crucial for elucidating their mechanisms of action in cardiac repair.

Although EV-based cardiac therapy has generally been deemed safe, recent concerns have emerged regarding possible pro-arrhythmogenic complications, which still require systematic evaluation. Nonetheless, recent evidence indicates that EV treatment should not significantly increase the risk of arrhythmia predisposition [95]. The optimization of EV-cardiac delivery and targeting is a rapidly evolving research field. While no adverse effects have been reported so far, further investigations should be performed to exclude any possible negative side effects in the long-term in terms of treatment safety and efficacy.

## 7. Conclusions

The emerging field of EVs as candidate therapeutics for CVD holds great promise. Surely, EVs represent very appealing biological entities with a huge impact. However, their translational potential still requires comprehensive investigation to establish them as reliable and effective medicinal products. On top of standardizing EV separation and concentration techniques, purification protocols should be optimized as well. Likewise, precise functional assessment of EV release and uptake should provide key insights to implement scientific knowledge, thereby accelerating progress in the regenerative field. 

## Figures and Tables

**Figure 1 ijms-25-06187-f001:**
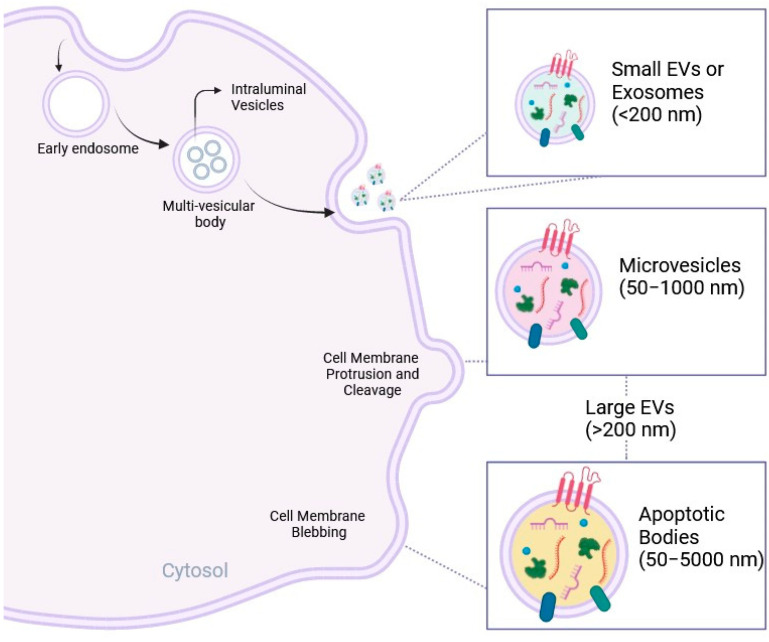
Schematic of EV population classification and biogenesis; *nm: nanometers*. Images have been produced using BioRender (www.biorender.com).

**Figure 2 ijms-25-06187-f002:**
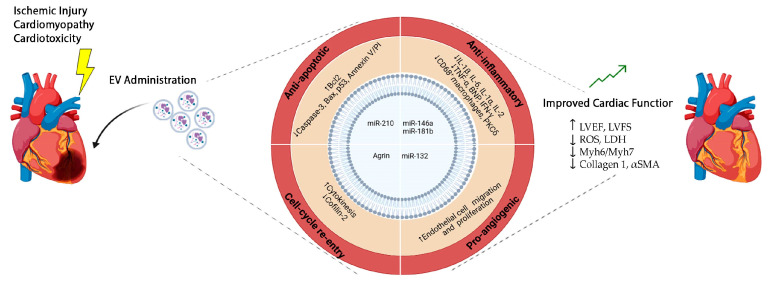
Schematic of progenitor cell EV paracrine effects and their putative molecular mechanisms of action against cardiac ischemic injury, cardiomyopathy, and cardiotoxicity, overall resulting in improved cardiac function. EV cardioactive paracrine potential includes the following: anti-apoptotic effects [71,72,73,74,75,76]; anti-inflammatory effects [72,73,74,76,77,78,79,80]; pro-angiogenic effects [75,81]; and stimulation of cardiomyocyte and cardiac stromal cell cell-cycle re-entry [82,83,84]. *miR: microRNA; LVEF: Left Ventricle Ejection Fraction; LVFS: Left Ventricle Fractional Shortening; ROS: Reactive Oxygen Species; LDH: Lactate dehydrogenase; Myh: Myosin Heavy Chain; aSMA: alpha-Smooth Muscle Actin.* Images have been produced using BioRender (www.biorender.com).

**Figure 3 ijms-25-06187-f003:**
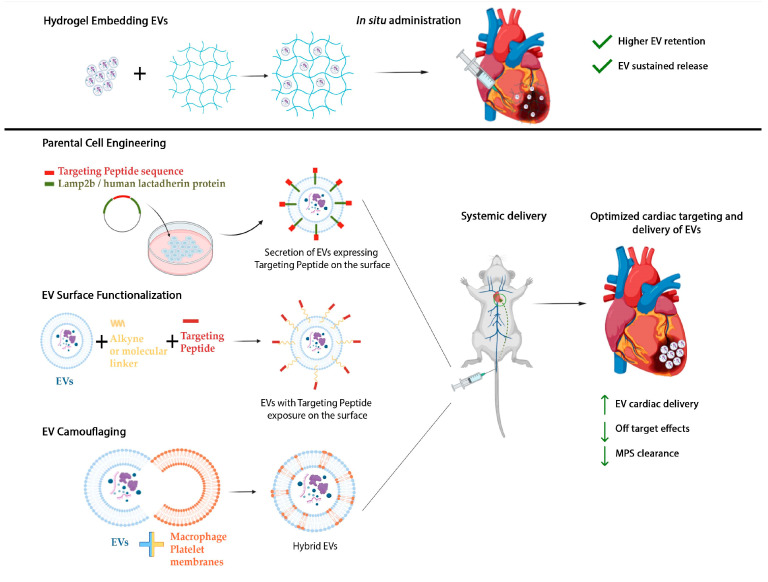
Schematic of the implementation of macro- and nano-application strategies to enhance long-term retention and controlled release of EVs (*Hydrogel Embedding EVs*), and to optimize specific delivery to the heart and cardiac targeting of therapeutic EVs (*Parental Cell Engineering; EV Surface Functionalization; EV Camouflaging*). *MPS: Mononuclear Phagocyte System.* Figures have been produced using BioRender (www.biorender.com).

**Table 1 ijms-25-06187-t001:** EV paracrine effects in preclinical models of cardiovascular disease.

Cell Source	EVs	In Vitro Outcome	In Vivo Outcome
**Adult** **MSCs**	hBM-MSC-EVs	Pro-angiogenic effect on HUVEC [81] **Model of myocardial oxidative stress:** ⬇ mNVCM Apoptosis [71]	**Rat model of AMI**: ⬇ Infarct Size Neovascularization ⬆ Cardiac Function [81]
	Rat BM-MSC-EVs	**Model of cardiac hypertrophy on H9c2 cells:** ⬇ Bax, Caspase-3 ⬆ Bcl-2 ⬇ BNP, TNF-α ⬇ IL-1β, IL-4, IL-6 [73,74]	
	Human AD-MSC-EVs	**Model of cardiac hypertrophy with iPSC-CM:** ⬇ ANF, COL1A1, IL-6 [77]	
	Murine AD-MSC-EVs		**Rat model of Doxorubicin-induced HF:** ⬆ ATP content, EF, FS ⬇ ANP, Bax, Caspase-3, p53 [85]
	Rat AD-MSC-EVs		**Rat model of AMI**: ⬆ Cardiac function M2 macrophage transition ⬇ IL-6, IL-1β, IFN-γ, TNF-α [77]
**Fetal/** **Perinatal MSCs**	Human AFSC-EVs		**Rat model of AMI**: ⬆ Cardiac function Cell-cycle re-entry Endogenous regenerative processes [73] **Rat model of ischemia/reperfusion:** ⬇ Infarct size [86] **Rat model of ISO-induced fibrosis:** ⬇ Collagen1, α-SMA ⬆ Angiogenesis [87] **Mouse model of AMI:** ⬆ Cell-cycle progression [83]
	Human UC-MSC-EVs		**Rat model of AMI:** ⬇ Cardiac fibrosis ⬆ Cardiac function [88]
	Human PMSC-EVs		**Mouse model of AMI:** ⬇ AST, BNP ⬇ IL-1β, IL-6 and TNF-α [78]
**Cardiac** **stromal cells**	Murine EPDCs-EVs	⬆ mNVCM proliferation **EHM cryoinjury model** ⬇ Contractile function [84]	**Mouse model of AMI:** Cell-cycle re-entry [84]
	Human CPC-EVs	**HL-1 serum deprivation:** ⬇ Cell death HUVEC tube formation [75]	**Rat and porcine model of AMI:** ⬇ Cell apoptosis ⬆ Angiogenesis [75,94,95] **Murine model of** **doxorubicin/trastuzumab-induced** **cardiotoxicity:** ⬇ ROS levels ⬇ Collagen1 deposition ⬇ Inflammation, reducing CD68+ macrophages [76]
**iPSCs**	Murine iPS-EVs		**Mouse model of AMI:** ⬆ Cardiac Function [99]
	Human iPSC-Pg-EVs		**Mouse model of chronic HF:** preserved LV function [100] **Mouse model of AMI:** ⬇ M1 macrophage ⬆ M2 macrophage ⬇ IL-1α, IL-2, IL-6 ⬆ IL-10 [80] **Rodent model of chemotherapy-induced cardiomyopathy:** ⬇ Maladaptive remodeling ⬇ Myh6/Myh7 [97]

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
