# Peer review of "Evolving Strategies for Extracellular Vesicles as Future Cardiac Therapeutics: From Macro- to Nano-Applications"

_ijms, 2024, doi:10.3390/ijms25116187_

Round 1
Reviewer 1 Report
Comments and Suggestions for Authors
The review article proposed by Guerricchio et al. brings together novel information on the application of EVs in cardiovascular research, a cutting-edge topic in the scientific community. Although the information is very dense, the manuscript is adequately written. I leave the following comments to the authors.
1. L72 versus L88. Once words are abbreviated for the first time, they should not be abbreviated again. On the contrary, if they are first time, they should be defined (L-203, L-213, L.214, among others).
2. The word "responder" (L131, L134) and "vai" (L514) are correct.
3. Many references are missing, to mention some examples, L-57-59, L-131-134, L149-153, L183-185, L222-225, among others. Please review the information and complement with the corresponding reference.
4. Regarding point 3. The authors could also describe the effect of drugs on VE release in patients with AMI (e.g., PMID: 36015073).
5. The paragraphs are extremely dense, so it would be good to divide them, to make the reading clearer.
6. In the sense of EVs released from different sites, the authors could provide a picture/table summarizing their function and effect in the different contexts they address throughout the manuscript, as this information is not covered in Figure 1.
7. Have EV cardiac delivery methods shown negative effects? It would be convenient to describe or add as study perspectives.
8. Review punctuation issues throughout the text.
Comments on the Quality of English LanguageMinor editing of English language required
Author Response
We would like to thank the Reviewer for their feedback.
We have addressed the comments with the required editing and the additional information as highlighted in yellow in the text.
Additional references have been provided in this revised version of the manuscript; please refer to pages 2, 3, 4, 5 and in the Reference section.
The effect of drugs on EV release in AMI patients has also been included in paragraph "3. Exploiting EVs in cardiovascular disease", page 4.
A table indicating the most relevant EV paracrine effects in preclinical models of cardiovascular disease (Table 1, pages 8-9 of the revised manuscript) has also been added to the text.
To the best of our knowledge, we are not aware of any significant negative effect of EV treatment by cardiac delivery. Discussion on putative side effects of EV cardiac-targeting has been mentioned as critical aspect for future perspective (page 15 in the revised manuscript).
English grammar and style have been revised by Dr. Antonietta Silini, from Fondazione Poliambulanza Istituto Ospedaliero in Brescia, Italy, who is a native English speaker (as indicated in the Acknowledgement section).
Reviewer 2 Report
Comments and Suggestions for Authors
Extracellular vesicles have received extensive attention in recent years due to their potential diagnostic and therapeutic potential. The authors present a well-organized and thorough review of studies focused on the therapeutic effects of extracellular vesicles in preclinical models of heart failure. While a number of reviews have been written regarding the cardiovascular effects of extracellular vesicles, or subgroups of extracellular vesicles, the present review is particularly insightful and will be of interest to readers of IJMS. Several relatively minor suggestions are presented below:
- - In general, the manuscript is well-written; however, careful grammatical editing is needed to more clearly communicate the information.
- - While numerous illustrations can be found in the literature depicting formation and characteristics of subgroups of extracellular vesicles, it would be beneficial to incorporate one in the manuscript to accompany the discussion in section 2.
- - There are several places that references should be incorporated including line 41 (billions of cardiomyocytes are irreversibly lost), line 151 (shown in different studies), line 190, line 224 (been also reported).
- - In the paragraph discussing purification methods of extracellular vesicles (lines 556-587), it would be helpful to briefly mention disadvantages to various purification methods.
Comments on the Quality of English LanguageThe English language needs minor editing.
Author Response
We would like to thank the Reviewer for their input and suggestions.
We have addressed the Reviewer's suggestions by providing additional information, as highlighted in yellow in the text of the revised manuscript.
We have now added a schematic picture (Revised Figure 1) summarizing EV subset classification and biogenesis (as embedded in the revised manuscript, page 4).
Additional references have been provided (page 1, 4 and 5 and Reference section, page 15-21).
Pros and cons of the different protocols and methods for EV separation and concentration have been included in the discussion in paragraph "6. Translational challenges" (page 14).
English grammar and style have been revised by Dr. Antonietta Silini, from Fondazione Poliambulanza Istituto Ospedaliero in Brescia, Italy, who is a native English speaker (as indicated in the Acknowledgement section).